# Overview of Antifungal Drugs against Paracoccidioidomycosis: How Do We Start, Where Are We, and Where Are We Going?

**DOI:** 10.3390/jof6040300

**Published:** 2020-11-19

**Authors:** Lívia do Carmo Silva, Amanda Alves de Oliveira, Dienny Rodrigues de Souza, Katheryne Lohany Barros Barbosa, Kleber Santiago Freitas e Silva, Marcos Antonio Batista Carvalho Júnior, Olívia Basso Rocha, Raisa Melo Lima, Thaynara Gonzaga Santos, Célia Maria de Almeida Soares, Maristela Pereira

**Affiliations:** 1Laboratory of Molecular Biology, Institute of Biological Sciences, Federal University of Goiás, Goiânia 74690-900, GO, Brazil; o.amanda.alves@gmail.com (A.A.d.O.); diennybiotec@gmail.com (D.R.d.S.); katherynelohany@gmail.com (K.L.B.B.); smallbinho@hotmail.com (K.S.F.eS.); carvalhojr.biolic@gmail.com (M.A.B.C.J.); oliviabassorocha@gmail.com (O.B.R.); raisamelolima@hotmail.com (R.M.L.); thaynara075@hotmail.com (T.G.S.); cmasoares@gmail.com (C.M.d.A.S.); 2Institute of Tropical Pathology and Public Health, Federal University of Goiás, Goiânia 74605-050, GO, Brazil

**Keywords:** Paracoccidioidomycosis, antifungal, nanotechnology, patent

## Abstract

Paracoccidioidomycosis is a neglected disease that causes economic and social impacts, mainly affecting people of certain social segments, such as rural workers. The limitations of antifungals, such as toxicity, drug interactions, restricted routes of administration, and the reduced bioavailability in target tissues, have become evident in clinical settings. These factors, added to the fact that Paracoccidioidomycosis (PCM) therapy is a long process, lasting from months to years, emphasize the need for the research and development of new molecules. Researchers have concentrated efforts on the identification of new compounds using numerous tools and targeting important proteins from *Paracoccidioides*, with the emphasis on enzymatic pathways absent in humans. This review aims to discuss the aspects related to the identification of compounds, methodologies, and perspectives when proposing new antifungal agents against PCM.

## 1. Introduction

The infection caused by fungi of the genus *Paracoccidioides* was first described by Adolpho Lutz in 1908. In 1971, during the meeting of several mycologists from Latin America in Medellín-Colombia, the term Paracoccidioidomycosis (PCM) was made official to designate the systemic granulomatous infection caused by the thermodimorphic fungi of the genus *Paracoccidioides* [1,2]. Several aspects of the infection remained unknown for a long period, including the taxonomic classification of the pathogen and a more appropriate therapeutic approach [3]. Nowadays, however, the importance of this disease in Latin America is recognized, with it being one of the main causes of deaths due to fungi infections [4].

The eradication of the fungus in the tissues is slow and treatment might last from months to years of antifungal administration. *Paracoccidioides* spp. are sensitive to various systemic antifungals, but the therapeutic options are limited to antifungals that act on two main targets, plasma membrane and folic acid synthesis [5]. Several new compounds with antifungal properties have been proposed against PCM over the last decade. Thus, this review focus on studies that have identified alternative compounds to the current treatment of PCM, as well as the strategies used for the development of new antifungal drugs. 

To review the antifungal compounds used in the treatment of PCM, a search was performed in SciELO (Scientific Electronic Library Online, https://scielo.org/), PubMed (https://pubmed.ncbi.nlm.nih.gov/), PubMed Central (https://www.ncbi.nlm.nih.gov/pmc/), and Virtual Health Library (https://bvsalud.org/) databases in August 2020. We used the Medical Subject Headings (MeSH) terms “Paracoccidioidomycosis” OR “Infection, *Paracoccidioides brasiliensis*” OR “South American Blastomycosis” OR “*Paracoccidioides brasiliensis* Infection” OR “Paracoccidioides” OR “*Paracoccidioides brasiliensis*” AND “Antifungal” OR “Agents, Antifungal” OR “Fungicides, Therapeutic” OR “Antibiotics, Antifungal” OR “Antifungal Antibiotics” and for searches in Portuguese we use the Health Sciences Descriptors (DeCS) terms: “*Blastomyces brasiliensis*” OR “*Paracoccidioides*” OR “Paracoccidioidomicose” OR “Blastomicose Sul-Americana” OR “Granuloma *Paracoccidioides*” OR “Infecção por *Blastomyces brasiliensis*” OR “Infecção por *Paracoccidioides brasiliensis*” AND “Antifúngicos” OR “Agentes Antifúngicos” OR “Agentes Antimicóticos” OR “Antimicóticos” OR “Fungicidas Terapêuticos”.

These results are presented in complementary tables consisting of antifungal derived from the synthetic and semi-synthetic compounds tested against *Paracoccidioides* spp. (Appendix A), the antifungal compounds from plants and microorganisms’ sources tested against *Paracoccidioides* spp. (Appendix A), and the nanoparticles of antifungal immobilized compounds tested against *Paracoccidioides* spp. (Appendix A).

## 2. Treatment of Paracoccidioidomycosis: An Overview

Figure 1 shows the historical context that defines the treatment of PCM currently. The first alternative treatment emerged in 1940, 32 years after the first reports of the disease by Lutz, using sulfapyridine, which is a sulfonamide derivative [6]. Sulfonamides are recognized for their broad spectrum of antibiotic activity, interrupting the growth of microorganisms by the competitive inhibition of the aminobenzoic acid (PABA) to the enzyme dihydropteroate synthase in the folate synthesis pathway. The latter is an essential component in the synthesis of nucleic acids and proteins [7,8]. Sulfa derivatives were also used in the treatment of PCM according to the severity of the disease, including compounds of low, moderate, and high excretion by the body [5].

Treatment with amphotericin B (AmB) was introduced by Lacaz and Sampaio [6] in 1958. This drug was established as an antifungal agent with a wide spectrum of action, acting mainly through the formation of complexes with ergosterol. These are molecules present in the membranes of fungal cells and responsible for the formation of transmembrane channels that allow the extravasation of cytoplasmic components [9]. It is indicated for severe and disseminated forms. Despite its effectiveness, AmB is also able to bind, to a lesser extent, to the cholesterol present in mammalian cells, producing several toxic effects on the host during the course of treatment, including acute symptoms such as nausea, vomiting, fever, hyper and hypotension, and hypoxia, in addition to chronic nephrotoxicity [10,11]. New formulations of AmB were developed for incorporation into liposomes, resulting in a better tissue distribution and less toxicity [5].

Another treatment option for PCM emerged with the introduction of cotrimoxazole (CMX) in 1973. This drug is a synergistic association between sulfamethoxazole (a sulfonamide derivative) and trimethoprim, and is used in patients with mild to moderate forms of PCM and neuroparacoccidioidomycosis [12,13,14]. CMX acts by inhibiting the enzymes involved in the synthesis of tetrahydrofolic acid, leading to the depletion of intracellular folate, which is essential for the growth of the pathogenic organism [15]. 

The development of azole derivatives certainly supported the expansion of the arsenal of antifungals against PCM. These compounds are the most common agents used in the treatment and prevention of a wide spectrum of mycoses, preventing the biosynthesis of ergosterol by inhibiting the CYP450-dependent enzyme, lanosterol 14-α-demethylase [16,17]. Two years after its publication in 1979, ketoconazole was introduced as an antifungal for the treatment of PCM [18,19,20]. Despite the success of this drug in controlling mild and moderate forms of the disease, its use is no longer recommended due to hepatotoxicity and adrenal insufficiency [21]. Thus, ketoconazole was eventually replaced by the first generation of triazoles, especially after the introduction of itraconazole (ITZ) in 1987 [22].

Some other alternative treatments have been proposed over the past two decades. A case report published in 2000 showed a patient who did not respond to the initial treatment with CMX but achieved clinical, mycological, and radiological cures two years after the end of treatment with terbinafine [23]. Terbinafine is an antifungal of the allylamine class that has a similar mechanism of action to azole agents, blocking the ergosterol biosynthesis pathway by inhibiting the squalene enzyme epoxidase [24]. A representative of the second-generation triazoles known as voriconazole has a similar efficacy to ITZ, and is useful in the treatment of neuro-PCM due to the greater penetration of this drug into the central nervous system compared to ITZ [25].

The publication of the consensus on PCM in 2006 allowed the creation of guidelines to formalize the PCM clinical treatment. Shikanai-Yasuda recommends the use of ITZ as the drug of choice for the treatment of mild and moderate forms of PCM, followed by CMX and AmB, according to the severity of the disease. The consensus also points to the possibility of using voriconazole, posaconazole, and isavuconazole to replace itraconazole, paying attention to costs, clinical evidence, and drug interactions [14].

Even after one hundred years of investigation into the disease, the therapeutic approaches face several issues. The main problem to overcome concerns the long period of treatment required by the currently available antifungals, which occasionally results in patients giving up therapy [26]. Another issue is the possibility of *Paracoccidioides* developing resistance against these antifungals. In *Paracoccidioides* spp., genes that were homologous to the cerebellar degeneration-related protein (CDR1, CDR2) and multi-drug resistance (MDR1) of *Candida albicans*, the pleiotropic drug resistance (PDR5) of *Saccharomyces cerevisiae*, and the ABC transporter genes of *Aspergillus* spp. were observed; all of them were related to resistance against azoles [27]. Thus, as the current treatment against PCM is mainly based on azole derivatives, these genes may play a similar role in *Paracoccidioides* spp., with the possibility of developing resistant isolates. In fact, the in vitro resistance of *Paracoccidioides* spp. was evidenced against azole derivatives [28]. Cermeño et al. carried out a sensitivity test of several species of *Paracoccidioides* spp. and observed the resistance against caspofungin (94.7%), followed by 5-flucytosine (52.6%) and AmB (47.4%) [29]. 

In addition to antifungal therapy, the treatment of possible sequelae of PCM such as pulmonary fibrosis and the prevention of opportunistic diseases should also be considered. An additional therapy proposal for PCM aimed at reducing pulmonary fibrosis is the combination of itraconazole-pentoxifylline. However, although the combination has shown promising results in mice, there are still no reports on testing in humans.

## 3. New Anti-*Paracoccidioides* Compounds: Do We Already Have Any Ideal Antifungals?

The ideal antifungal agent should have a broad activity; be selective for fungal targets; and have reduced adverse effects, limited interactions, many administration routes, and low resistance. Thus, proteins belonging to the metabolic pathways involved in amino acid metabolism, cell wall, ergosterol biosynthesis, response to oxidative stress, and alternative sources of carbon are targets for antifungals and explored in the process of identifying new selective compounds. Such parameters are essential for the survival of the pathogen and additionally those targets should be absent in humans [30]. Over the years, several techniques and approaches helped to search for promising antifungals (Figure 2).

The ability of a compound to inhibit fungal growth is known as Minimum Inhibitory Concentration (MIC). MIC was first used in a PCM study in 1982 to assess the in vitro effects of CMX against *P. brasiliensis* clinical isolates [31]. However, one of the major problems is the lack of standardization in these tests, which is directly related to the variety of MIC protocols found in the literature. Experimentally, tests that identify MIC values have been performed for several years through the macrodilution technique or by counting colony-forming units. This technique was improved by converting these assays to microdilution using 96-well microplates in 2003 [32]. In addition, the proposition of using resazurin dye as a marker of metabolic activity provided even greater practicality for those assays in 2013 [33]. Studies reported that certain variables, such as culture medium and incubation time, have a major impact on the results [34]. Although there is a trend towards the use of the RPMI-1640 culture medium, as recommended by the Clinical and Laboratory Standards Institute (CLSI) [35], the incubation time still remains undefined, ranging from 3 to 15 days of incubation [36,37].

The search for new antifungal candidates has been driven by synthetic, semi-synthetic, plant. and microorganism compounds. In this review, we describe some of the compounds tested against *Paracoccidioides* spp., and additional compounds are listed in Appendix A.

Potential compounds act as growth inhibitors against *Paracoccidioides* spp. in low concentrations (Appendix A), such as alkyl gallates (0.004–16 µg/mL), an N-Glycosylation inhibitor [38,39]; the thiosemicarbazone lapachol, which acts on the plasma membrane (0.01–0.1 µM) [40]; azasterol analogs (0.5–10 µM) [41] and hydrazone derivatives (0.1–5 µM) [42], as inhibitors of ergosterol biosynthesis. These compounds are less potent when compared to azole derivatives, such as itraconazole with a 0.003–0.05 µM MIC [33]; luliconazole, a topical antifungal repositioned against *Paracoccidioides* spp. through in vitro assays and with an MIC ranging from 0.0005 to 0.0007 µM [43]; and butaconazole, which is used for the local treatment of vulvovaginal candidiasis and also repositioned against *Paracoccidioides* spp. with an MIC of 0.001–0.002 µM. Both compounds are inhibitors of the ergosterol biosynthesis, indicating that this molecular target is very promising. Antimicrobial peptides have also been proposed as potent inhibitors of *Paracoccidioides* spp., such as MK58911, a peptide analogue of mastoparan, which presented an MIC of 7.8 μg/mL against *P. brasiliensis* and 15.6 μg/mL against *P. lutzii* [44]. Lactoferrin-derived peptides presented MIC values between 0.63 and 1.25 μg/mL [45].

The medicinal potential of plants as antifungals comes from their extracts, essential oils, and chemical constituents [46]. Several studies have reported compounds with antifungal properties against PCM, such as argentilactone derived from *Hyptis ovalifolia* [47], oenothein B derived from *Eugenia uniflora* [48,49], fatty acid methyl esters and compounds derived from *Annona cornifolia* essential oils [50], hydroalcoholic extracts from the species *Piper regnellii* and *Baccharis dracunculifolia* [51,52], and curcumin from *Curcuma longa* [53]. Another natural compound is ajoene, derived from *Allium sativum*, which exerts inhibitory effects against morphological transition and in yeast cells, with possible involvement in the sulfhydryl metabolism of *P. brasiliensis* [54].

Exploring the universe of natural source compounds, microorganism-derived compounds were tested against *Paracoccidioides* spp. (Appendix A). The extracts and cytochalasin E isolated from *Aspergillus felis* presented an MIC of 31.2 µg/mL (3.6 µM) [55]. The extract of the endophytic fungus *Fusarium* sp. containing T2-toxin was able to inhibit clinical strains of *P. brasiliensis* with an MIC ranging between 75 and 640 nM and the extract containing 8-n-butyrylneosolaniol and 8-isobutyrylsolaniol had MIC values from 160 to 640 nM [56]. Altenusin, isolated from the endophytic fungus *Alternaria* sp., exhibited an MIC between 1.9 and 31.2 μg/mL [57]. Another important compound is farnesol, a *C. albicans* quorum sensing molecule, which acts as a potent antifungal inhibiting the growth and dimorphism in μM concentrations [58].

## 4. Compounds with Fungicidal Activity 

The establishment of PCM depends on two fundamental factors—the virulence of the infectious agent and the host’s immune response capacity. After infection, the fungus can remain latent in the host without causing symptoms and the control of fungal replication is dependent on an effective immune response with the activation of CD4 and CD8 T lymphocytes and macrophages [59]. However, fungal cells persist inside granulomas, and when the individual for some reason has a compromised immune system the infection spreads throughout the body, instilling the most severe form that evolves rapidly and with a higher mortality rate [14]. 

Some definitions are attributed to antifungals, considering the effect on the fungus. Thus, drugs that inhibit fungal growth are considered fungistatic, while drugs that actually cause death are considered fungicides [60]. Studies comparing treatment in co-infected patients with patients only infected with the fungus showed that the recurrence of PCM was greater in patients with HIV, because the antifungal drugs only inhibited the growth of the fungus once the compromised immune system was unable to fight the fungus [61]. Among the therapeutic arsenal used in the treatment of PCM, the only drug reported with fungicidal activity in *Paracoccidioides* spp. is AmB, which is indicated in the most severe cases of the disease at a dose of 1 mg/kg/day [14]. Due to its low selectivity with systemic toxicity, its antifungal action is overshadowed by serious potentially fatal side effects [62,63].

Experimentally, fungicide concentration tests are used in order to verify the ability of some compounds to promote the effective death of the fungus. The protocol consists of cultivating samples at concentrations above MIC values, in solid culture medium, with the subsequent incubation and visual verification of the fungal growth or inhibition. The minimum fungicidal concentration (MFC) is defined as the lowest concentration of the compound in which the growth of the fungus is not visualized [43]. The technique for defining the fungicidal action of compounds in the fungi of the genus *Paracoccidioides* date from 1984 [64], when some antimycotic drugs, such as AmB, ketoconazole, and rifampicin, were evaluated against strains of *P. brasiliensis* where the MFC values were between 0.36 and 500 μg/mL. 

Some compounds have their MFC values described in Table 1. In general, the compounds are promising, with an acceptable fungicidal potential and selectivity rate. Although the technique for determining the MFC is easy and fast, it is not universally applied.

## 5. Evaluation of Synergistic Effects of New Compounds Associated with Traditional Antifungals

The performance of the clinical antifungals depends on the immunological and physiological status of the patient, hypersensitivity reaction, and pharmacokinetics of such drugs. Combined antifungal therapy (CAT) is an alternative to monotherapy because it increases the spectrum of their activity through the association of antifungals [73]. This approach explores the synergistic interaction of antifungals, and the association might exceed any individual contribution of each drug [74]. Thus, some objectives of CAT are to increase the spectrum of action, reduce the development of fungal resistance, reduce toxicity, and increase fungicidal activity. On the other hand, the association of antifungals might result in antagonistic interactions when the effect of the association is less than the individual effect of each antifungal or results in indifferent interactions when the association is not significant or absent [75,76].

CAT was introduced in the treatment of cryptococcal meningitis in the 1970s [77] and is currently described for the treatment of several fungal infections, including candidiasis [78], aspergillosis [79], and fusariomycosis [80]. However, in the treatment of PCM, this approach is used only for CMX, which is the result of the synergistic association of sulfamethoxazole and trimethoprim [14]. This indication comes from studies carried out in 1982, which reported that the most synergistic effects take place in clinical isolates of *P. brasiliensis* treated with sulfamethoxazole/trimethoprim in a ratio of 5:1 [31].

The major challenges of the combinatorial therapy are cost and time. Several methodologies evaluate the interaction among antifungal compounds, such as the checkerboard test. This technique combines several concentrations of two compounds arranged in an orderly manner on a microplate. The concentrations of compound (A) decrease in the vertical direction and the concentrations of compound (B) decrease in the horizontal direction, with subsequent incubation with the microorganism under study. The MIC values of the individual compounds are determined in order to identify which compound has the best result. The effect of the interaction results from the Fractional Inhibitory Concentration Index (FICI). It is calculated by the formula FICs = FIC_A_ + FIC_B_, where FIC_A_ is the MIC of compound A combined with B and then divided by the MIC of compound A alone; FIC_B_ is the MIC of compound B combined with A and then divided by the MIC of compound B alone. The FICIs are interpreted as synergic for FIC ≤ 0.5, additive for FIC > 0.5 to ≤1, indifferent effect for FIC > 1 to ≤4, or antagonism for FIC > 4 [81,82].

The association between commercial antifungals and natural and synthetic compounds against *Paracoccidioides* spp. showed an increase in the potency of the antifungal effect, the association also resulted in a reduction in the concentration of the compounds and the consequent reduction in their toxicity. The synthetic compound CaCS02, an inhibitor of the enzyme chorismate synthase, was identified by bioinformatics tools and demonstrated the synergistic association of CaCS02 with AmB. This interaction reduced the MIC of AmB fourfold (from 0.5 to 0.125 μg/mL) and reduced the MIC of CaCS02 eightfold (from 32 to 4 μg/mL) [65]. The oxadiazole compounds LMM5 and LMM11 showed an additive effect when associated with AmB, resulting in a reduction in the MIC value of AmB from 32 to 4 μg/mL, and the MIC of oxadiazole compounds reduced from 16 to 4 μg/mL [67].

Another study evaluated 17 synthesized compounds derived from 4-methoxynaphthalene-N-acylhydrazones. The compounds 4a-c, 4g, and 4k had synergistic activity with AmB against *P. brasiliensis*. In addition, the compound 4K was identified as a potential candidate for antifungal therapy in patients co-infected with *P. brasiliensis* and *Mycobacterium tuberculosis* [83].

The synergistic association between AmB and copaiba oil was demonstrated in *Paracoccidioides* spp. and resulted in a reduction in the MIC values from 0.75 to 0.18 μg/mL and from 62.5 to 31.25 μg/mL, respectively [37]. Quinolinyl N-oxide chalcones 3 and AmB or CMX showed synergistic effect and reduced the MIC value of Amphotericin from 0.81 to 0.20 μM and CXM from 9.5 to 0.5 mM [84]. ITZ and synthetic compounds inhibitors of the enzyme homoserine dehydrogenase (HS1 and HS2) showed an additive effect against *P. brasiliensis* and *P. lutzii*; the ITZ MIC reduced from 0.008 to 0.001 [85].

Antifungals can interact directly with phagocytic cells (monocytes/macrophages and neutrophils). This interaction might promote synergistic and antagonistic effects during antifungal therapy. The exposure of immune cells to drugs leads to the activation of membrane receptors, culminating in a positive regulation of pro-inflammatory responses [86]. ArtinM, extracted from the seeds of *Artocarpus heterophyllus*, showed immunomodulatory activity, inducing IL-12 production and Th1-type immune response polarization in rats infected with *P. brasiliensis*. This last approach resulted in a reduced fungal load [87].

Antifungal drugs induce immunomodulatory activity, such as AmB, which interferes with the secretion of pro-inflammatory chemokines and cytokines [88]. An exacerbated increase in the host’s inflammatory response results in tissue damage and further worsens the patient’s health. The concomitant use of antifungal therapy with corticosteroids in patients with severe forms of PCM results in clinical improvement and prevent complications [89].

The additive effect of the association of chemotherapy with immunizing peptides reduces the treatment time and improves the prognosis of anergic cases [90,91]. The interaction analysis is a promising tool for screening synergistic or antagonistic interactions and should be applied in the identification of new compounds. Thus, combinatorial antifungal therapy is a potential alternative for the treatment of PCM and other invasive infections.

## 6. Computational Methods: A Potential and Strategic Way to Identify Novel Antifungals

The development of new drugs is a complex and expensive process that requires long time and high investment from pharmaceutical industries [92]. The computational approach plays an important role in drug research and development, reduces time and cost by filtering large libraries of compounds into smaller sets that can be tested experimentally, and guides the optimization of the affinity and pharmacokinetic properties of the selected compounds [93]. 

The development of new approaches could enable a more efficient and rapid treatment and fewer side effects. The choice of therapeutic targets that are absent in humans may result in less toxic effects [94]. Thus, docking and molecular dynamics approaches for the identification of new promising compounds against PCM have been proposed since 2014, exploring the interaction between argentilactone and isocitrate lyase based on molecular dynamics and docking [95]. 

Costa et al. performed a virtual screening analysis of natural compounds in 2015. This approach used the ZINC database -http://zinc15.docking.org- which contains more than 89 thousand active compounds against the enzyme malate synthase of *Paracoccidioides* spp. [96]. Promising inhibitors compounds against the enzyme isocitrate lyase were found in a virtual screening approach using the AutoDock Vina program and the ZINC databank. The virtual screening was performed under two conditions (with and without Mg^2+^) and, among the compounds tested in vitro, the most promising one with antifungal activity and a 100% inhibition of the enzyme activity was from ligands selected from isocitrate lyase bound to Mg^+^ [66]. This approach points to the importance of considering enzyme cofactors for the computational analysis, minimizing the conditions used in experimental tests.

Abadio et al. [68] screened compounds from Life Chemicals -https://lifechemicals.com- with the ability to bind to the enzyme thioredoxin. Out of the 12 compounds selected for the in vitro tests, three showed antifungal activity against *Paracoccidioides* spp. and, among these, two inhibited the recombinant thioredoxin. Molecular docking simulations identified potential inhibitors of the enzyme homoserine dehydrogenase (HSD) from *P. brasiliensis*. Four compounds were selected to be tested in vitro [85]. Rodrigues-Vendramini et al. used homology modeling, virtual screening, and molecular dynamics to analyze the *P. brasiliensis* chorismate synthase (AutoDock and Molegro [97]) in order to identify new potential inhibitors against this enzyme. The study identified of a new compound with potent antifungal activity and that was able to reduce pulmonary inflammation [98]. 

The shape-based screening methodology identified new anti-*Paracoccidioides* compounds in 2018. Compounds that share similarity in molecular shape and volume could also share similarity regarding the biological activity. Hence, the virtual screening of chalcone-specific pharmacophores with known antifungal activity was carried out. This approach allowed the selection of 31 compounds with antifungal activity and with an additive effect in combination with AmB [84].

Drug repurposing is a strategy for identifying new therapeutic uses for approved or investigational drugs that are outside the scope of the original medical indication [99,100]. In 2019, two studies applied the drug repositioning method to suggest compounds for the treatment of PCM. In the first, researchers used as scaffold compounds with antifungal activity already described for *C. albicans* (LMM5 and LMM11) [101]. After applying the Tanimoto coefficient and molecular docking, the antiviral raltegravir was identified as a potent inhibitor of *Paracoccidioides* spp. [102].

A different approach applied data from the genome of *Paracoccidioides* spp. to identify orthologous proteins within data banks of compounds [43]. According to this methodology, proteins with a high similarity may share the same ligands. Hence, they performed a screening of orthologous proteins belonging to three isolates of *Paracoccidioides* spp. against two drug banks (DrugBank and Therapeutic Targets Database). After the comparison of essential *S. cerevisiae* genes and *Paracoccidioides* genes we performed homology modeling and molecular docking. Several commercially approved compounds or compounds undergoing clinical trials were selected for the in vitro tests, such as mebendazole, an anti-helminthic drug that has been proposed as a treatment against mycosis caused by *Cryptococcus neoformans* [103]; azoles, such as sertaconazole, butaconazole, and luliconazole and inhibitors of PbTOR2, such as BGT-226, vistusertib and dactolisib. 

A perspective for the identification of new anti-*Paracoccidioides* compounds is the use of computational methods of the type QSAR (Quantitative Structure–Activity Relationship), which is based on mathematical models that predict the biological activity of compounds from the relationships established between chemical structure and biological activity [104,105]. This bioinformatics tool enables the prediction of activity, properties, and toxicity of new compounds [106]. The construction of predictive models is highly dependent on the size, quality, and diversity of the data [107]. The limited number of compounds with biological activity known against *Paracoccidioides* spp. and the lack of standardization of MIC assays have limited the use of this methodology computational, which requires as input data MIC values and a large number of compounds [105].

## 7. The Identification of the Mode of Action of Compounds Based on Ohmics Tools

There is an increasing demand for new techniques to unravel the fungal pathogenesis, host–pathogen interplay, and eventually new treatment approaches. Genomics [108], transcriptomics [109], proteomics [110], chemoproteomics [111], and metabolomics [112] contribute to the discovery of new promising compounds with antifungal properties. The host-pathogen interaction leads the onset of the PCM infection [113] and the host and fungal responses drive the investigation of new strategies to combat the disease. Proteomic approaches provide insight into the *Paracoccidioides* infection on host metabolism, such as the immune response and the pathways involved in fighting the infection, and are able to identify targets in order to intensify defense mechanisms [114]. They also provide insight into the metabolism of the pathogen, pointing to novel virulence factors, mechanisms of action of fungal proteins, and new anti-virulence strategies to improve the treatment of the disease [115].

The first transcriptomic study regarding fungal response in the presence of antifungal prototypes shed some light on how such compounds may interfere with the pathogen survivability within the host and the results confirmed that the compound alters *Paracoccidioides* cell morphology [48,49]. Although they evidenced that the mode of action of oenothein B is based on the disruption of the cell wall, several other genes related to metabolism, energy, morphology, transcription, and transport were differentially expressed [48] and could potentially lead to new targets.

ITZ has been used to treat several fungal infections, including PCM. The mechanism of action of azole antifungal agents is based on the inhibition of ergosterol, which presents roles in the regulation of membrane structure and vesicle formation [116]. The transcriptomic response of *Paracoccidioides* cells treated with ITZ revealed that genes involved in ergosterol production, cellular transport, energy, transcription, cell rescue, defense, and virulence were differentially expressed [117]. The mode of action of ITZ on *Paracoccidioides* cells upregulates acetyl synthesis through several pathways, mainly beta-oxidation and amino acid degradation, and drives those molecules to ergosterol production. This same pattern has been found in the ohmic approaches of ITZ on other fungi species, such as *Saccharomyces cerevisiae* [118] and *C. albicans* [119]. Interestingly, ITZ drives a detoxification response via the increased expression of glutathione S-transferases and heat shock proteins (HSPs) in *Paracoccidioides*.

The mode of action of argentilactone against that pathogen was investigated in 2015, 2016, and 2018 by proteomics [120], transcriptomics [115], and chemoproteomics [47], respectively. Some common results of those three techniques can be evidenced. The proteomic and transcriptional profile of *Paracoccidioides* in response to the antifungal argentilactone showed an increase in the levels of stress-related proteins, such as the HSPs. In addition, the argentilactone chemoproteomic approach identified that this natural compound interacts with several HSPs [47]. These results, identified by ohmic studies regarding the mode of action of the new antifungal, suggest that *Paracoccidioides* HSPs are good targets of stress-inducing compounds (Figure 3).

The enzyme superoxide dismutase (SOD) presented increased levels in the proteomic and transcriptional profiles and interacted with the compound in the chemoproteomic approach of *Paracoccidioides* in the presence of argentilactone [47,115,120]. Thus, it is wise to state that the compound main mode of action is via increasing the stress conditions in the environment where the pathogen cells struggle to survive during the infection. Increasing the expression of SOD seems to be the first antioxidant defense against reactive species in *Paracoccidioides* spp. [121].

The proteomic profile indicated a remodeling of the carbohydrate metabolism in *Paracoccidioides* cells treated with argentilactone through a reduced expression of glycolytic enzymes [120]. The transcriptional profile identified that carbon metabolism is inhibited by argentilactone [115] and the chemoproteomic approach showed that the natural compound interacts with GAPDH, influencing the obtainment of energy by the fungal cells [47]. Interestingly, the proteomic results pointed that glucose consumption was reduced by the action of argentilactone and thus, glycolysis was partially blocked [120]. Thus, the ohmic techniques were complementary and together defined the mode of action of these compounds in *Paracoccidioides* spp.

The same path was taken to identify the mode of action of thiosemicarbazide (TSC). The transcriptomics results of *Paracoccidioides* cells treated with TSC showed that the compound induced the production of reactive species that lead to disruption of the mitochondrial membrane integrity. The immediate consequence for the pathogen cells under such condition was the impairment of TCA, fatty acid oxidation urea cycle, and amino acid metabolism [119,120].

The proteomic profile of *Paracoccidioides* under TSC treatment identified that the TSC mode of action occurs by the reduction in pyruvate and acetyl-CoA production, besides the reduction in glycolytic and TCA protein/gene levels. Thus, energy generation is severely impaired in the presence of TSC. Interestingly, proteins from the electron transport chain showed increased expression and, conversely, proteins related to respiration showed reduced levels of expression. This leads to the accumulation of electrochemical energy in the mitochondrion membrane and a consequent increase in reactive species. Eventually, ATP production is blocked in the pathogen cells treated with TSC [114].

TSC increases the levels of such species in *Paracoccidioides* [122].The mode of action of classic antifungals—azoles, for example—also features the production of reactive species [123]. TSC was found to be interacting with HSPs in the chemoproteomic approach and in both approaches, proteomic and transcriptomic, HSPs were found with reduced levels of expression, indicating that cells treated with TSC do not respond well to stress conditions. Thus, the results of proteomics, transcriptomics and chemoproteomic are similar and have driven the identification of its mode of action. All these results combined pointed to important protein targets of TSC.

## 8. Models for In Vivo Experiments

Animal models have been largely used, and they contribute with invaluable information regarding scientific knowledge. Preclinical investigations with the aid of such models have increased the understanding of most diseases that affect human beings. Several animal models have been successfully used either to test the response of the pathogen in the presence of classical antifungals or to predict the pathogen and host responses to new prototype drugs against PCM [124].

Regarding classic antifungal drugs, sulfonamides was proved to be effective in a guinea pig model against PCM [125]. Ketoconazole was successfully tested in mice, rats, hamsters, and guinea pigs [126], reducing the severity of lesions caused by the pathogen. The latter antifungal was more effective in guinea pigs and rats than in hamsters [127]. ITZ was tested in mice, rats, and guinea pigs, and it was more effective than fluconazole [126,128]. Saperconazole was also tested in animal models and in humans [129]. In another murine evaluation, the natural compound ajoene was effective in regulating *Paracoccidioides* cells growth [130], and the association of ajoene with sulfamethoxazole/trimethoprim was even more effective regarding the reduction in fungal burden [131]. Biochemical features were assessed in mice models of *Paracoccidioides* under treatment with the compound canova, which is a natural compound extracted from *Aconitum napellus* and *Arsenicum album*. Mice treated with canova developed fewer functional alterations in organs affected by the pathogen [132].

There is no doubt that mice are the most used model for testing PCM. However, ethical issues have limited their use and invertebrate animals have been suggested as a viable alternative to decrease the use of mammals in experiments due to ease of manipulation, low cost, and efficiency in the time employed [133]. A new animal model tested for *Paracoccidioides* pathogenesis was *Caenorhabditis elegans*. *C. elegans* in contact with *Paracoccidioides* was able to stimulate the expression of antimicrobial peptide genes. A different approach tested a synthetic peptide originated from *Pichia anomala* killer toxin. The antifungal activity of the peptide was assessed in mice and the results were evaluated by immunohistochemistry, showing that animals treated with the peptide presented fewer severe lesions and no fungal cells were found in the lungs of the hosts. The study concluded that such antimicrobial peptides could be applied as an adjuvant to the PCM regular therapy in order to decrease the period of treatment, which normally reaches years [134].

*Galleria mellonella* is the alternative model most often employed [135,136], since the innate immune response exhibits many structural and functional similarities when compared to the mammalian innate immune response [137]. Not many antifungals have been tested in this model against PCM so far, but the number of studies has increased over the last years and the results have led to a better understanding of the host–pathogen interaction and their response to new prototype drugs. ITZ and AmB were used in a *G. mellonella* model to assess the efficacy and toxicity of these classical drugs. They showed that both antifungals were effective leading to a higher rate of larval survival and reduced fungal load in the host [113]. Four peptides derived from phage and with anti-adhesive properties were tested for the ability to protect *G. mellonella* from infection by *Paracoccidioides* spp. The authors showed that these peptides increased the survival of larvae infected with *P. brasiliensis* and *P. lutzii* and suggested that anti-adherence therapy may be an important strategy to prevent PCM [138].

## 9. The Important Role of Nanotechnology in the Optimization of Antifungals

The clinical effectiveness of a drug depends on the bioactive properties and its local bioavailability. Several nanotechnological applications, including the use of nanoparticles, nanoemulsions, and nanocapsules have been used in the pharmaceutical areas in order to increase the efficiency of bioactive compounds and reduce their therapeutic doses [139].

Classic antifungals used in the treatment of PCM are the targets of optimization projects through nanotechnology (Appendix A). ITZ was encapsulated in polymeric nanoparticles composed of poly (D, L-lactide-co-glycolide) (PLGA), which provided a change in its contact surface, altered its bioavailability, and allowed the drug to act as a fungicide in a reduced dose, thus increasing its effectiveness. The initial MIC of ITZ-NANO was 6.25 µg/mL, however its efficiency was increased after 30 days with an MIC value of 0.8 µg/mL and after 60 days with an MIC value of 0.4 µg/mL [140].

Since 1980, lipid-based formulations of AmB have been developed in order to decrease toxicity. The main formulations tested were liposomal AmB, AmB lipid complex, and colloidal AmB dispersion [141]. Hence, new methodological approaches using nanotechnology have been developed with the aim of proposing new formulations of AmB. A formulation of poly(lactide-co-glycolide) (PLGA) or poly(lactide-co-glycolide)-poly(ethylene glycol) (PLGA-PEG) blend nanoparticles containing AmB was developed and resulted in greater antifungal action, decreased nephrotoxicity, and increased bioavailability [142]. In addition, it was able to decrease the number of administrations of the compound in in vivo models [143]. A nanocomplex comprising AmB loaded on the surface of magnetite nanoparticles pre-coated with a double layer of lauric acid exhibited potential antifungal activity with low cytotoxicity and no cytotoxicity to human urinary cells [144].

The association of dodecyl gallate with a nanostructured lipid system (NLS) showed promising antifungal activity and low toxicity in lung fibroblasts and zebrafish embryos [145]. A nanoemulsion composed of 10% oil phase (cholesterol), 10% surfactant (castor oil, polyoxyl60/PEG-hydrogenated, phosphatidylcholine), and sodium oleate was developed with the purpose of potentiating the effects of the antifungal 2-hydroxychalcone. The results were promising, with a high antifungal activity and selective index value. Furthermore, no toxicity was observed in the MIC concentrations in in vivo experiments [146]. Dodecyl protocatechuate, derived from an acid present in fruits, nuts, and vegetables, was conjugated with NLS and increased its antifungal action against *P. brasiliensis* and *P. lutzii* and its antioxidant activity. In addition, it was able to decrease its dosage and side effects [147]. Another natural compound which had reduced cytotoxic activity and increased solubility was *Copaifera langsdorffii* oil in the form of nanoemulsion [37].

## 10. What Is the Fate of the Identified Compounds? A Patent Review

Despite several published articles indicating new potent compounds, no new drugs have been added to PCM’s therapeutic arsenal in recent years. Therefore, in order to understand where these compounds described in the literature are being directed, we also reviewed the number of existing patents for the treatment of PCM. The search was carried out between July and September 2020 in patent banks, such as INPI-www.inpi.gov.br, Latipat-lp.espacenet.com, Espacenet-worldwide.espacenet.com, and Patentscope-https://patentscope.wipo.int/search/pt/search.jsf, with the following terms “Paracoccidioidomycosis” and “*Paracoccidioides* spp.” in Portuguese, Spanish, and English.

Several patents suggested compounds and peptide as new forms of PCM treatment (Appendix A). Vaccine patents and diagnostic tests, even when produced specifically for *Paracoccidioides* spp., were not taken into consideration. Patents with the same title but with different deposit numbers and/or publication dates were considered only once. Some patents were not produced specifically for the PCM treatment; however, as the patent authors suggested that their products could be used to combat fungi, including *Paracoccidioides* spp., these patents were considered.

We found 74 patents deposited by universities, mainly from Brazil and pharmaceutical companies. The patents encompass new synthetic compounds derived from natural compounds, compounds in nanoparticle formulations, peptides, and injectable pharmaceutical compositions. Although some of the patents are not specific to *Paracoccidioides* spp., the inventors of the patents suggested their use in the treatment of PCM, such as nikkomycin Z, a modified-nucleoside analog and inhibitor of the chitin synthesis of dimorphic fungal pathogens. Thus, since the cell wall is an important virulence factor of *Paracoccidioides* spp. and absent in human cells, this compound can be a therapeutic alternative in the treatment of PCM.

Despite the expressive number of patents filed over the last years for the treatment of PCM, these patents do not reflect the compounds suggested in the literature. In addition, few patents filed by the pharmaceutical industry cited PCM as their main focus, however these compounds and formulations could be an alternative for the treatment of PCM.

## 11. Conclusions

Given the importance of the development of antifungals against PCM, due to the lack of ideal therapeutic options this review shows that there are still major challenges to be overcome between the discovery of new compounds and the development of antifungals that could be used clinically. However, some researchers have proposed effective drugs with reduced side effects for humans. More research should be carried out in order to achieve treatments that decrease the risk of possible sequelaes of PCM, such as pulmonary fibrosis. In addition, it is necessary to fill the gap between the scientific publication of the biological activity of compounds and the continuation of research in the next stages of the drug development process, such as clinical tests.

## Figures and Tables

**Figure 1 jof-06-00300-f001:**
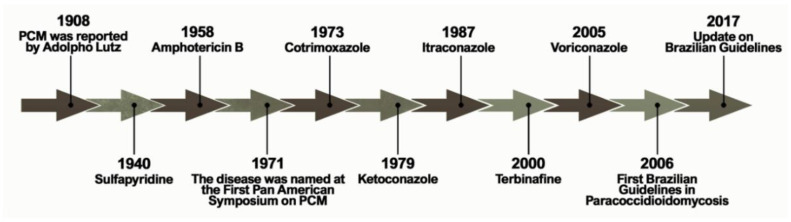
Important events related to the treatment and establishment of a consensus on Paracoccidioidomycosis.

**Figure 2 jof-06-00300-f002:**
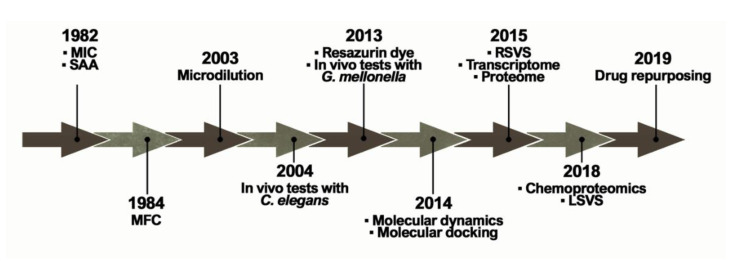
Methodologies used in the search and evaluation of anti-paracoccidioidomycosis compounds. MIC (Minimal Inhibitory Concentration), SAA (Synergistic Activity Assay), MFC (Minimal Fungicidal Activity), RSVS (Receptor shape-based Virtual Screening), LSVS (Ligand shape-based Virtual Screening).

**Figure 3 jof-06-00300-f003:**
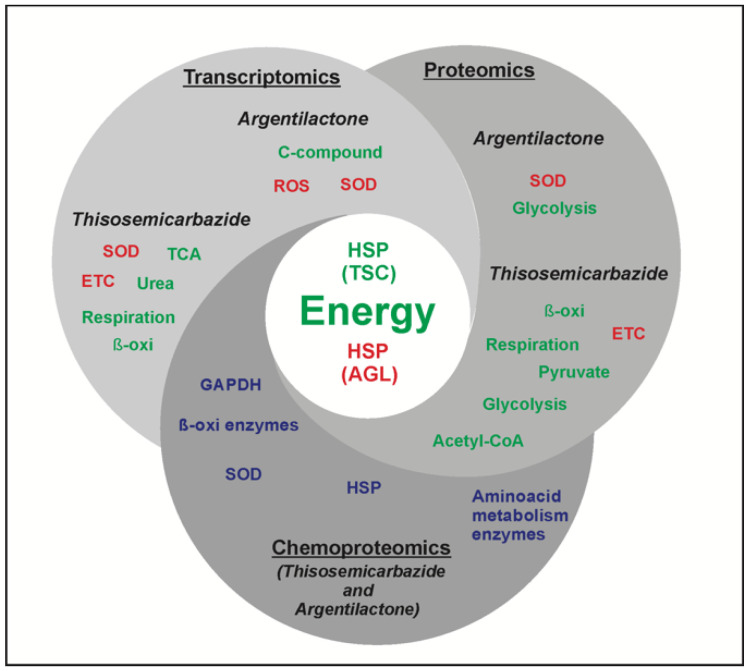
Modes of action of argentilactone and camphene thiosemicarbazide in *Paracoccidioides*. The diagram shows the two natural compounds that have been tested in *Paracoccidioides* cells according to three different ohmic approaches (proteomics, transcriptomics, and chemoproteomics). The central circle shows the common ways both compounds act in the pathogen; both restrict the ability of cells to produce energy through several pathways and, regarding stress conditions, the difference is that while thiosemicarbazide leads to a reduced HSP levels, argentilactone acts in such a way that protein/gene levels are increased. The green color indicates downregulation and the red color indicates upregulation. SOD—superoxide dismutase; ROS—reactive oxygen species; TCA—tricarboxylic acid; ETC—electron transport chain; GAPDH—glyceraldehyde-3-phosphate dehydrogenase.

**Table 1 jof-06-00300-t001:** Most potent fungicidal compounds.

Compounds	Target	MFC	Ref.
Identified by virtual screening			
CaCS02	Chorismate synthase	32 µg/mL	[65]
ZINC4559339	Isocitrate lyase	7.3–15.6 µg/mL	[66]
Oxadiazol LMM5	Thioredoxin reductase	1–32 μg/mL	[67]
Oxadiazol LMM11	Thioredoxin reductase	8–16 μg/mL	[67]
F3307-0100	Thioredoxin reductase	16.9 µM	[68]
HS7 (Zinc15967722)	Homoserine dehydrogenase	32 μg/mL	[69]
HS9 (Zinc2123137)	Homoserine dehydrogenase	8 μg/mL	[69]
Identified by drug repositioning			
Vistusertib	Phosphatidylinositol 3-kinase TOR2	1.0–4.2 μM	[43]
Mebendazole	Tubulin beta chain	13.2–26.4 μM	[43]
Butoconazole	Lanosterol 14-alpha demethylase	0.001–0.002 µM	[43]
Luliconazole	Lanosterol 14-alpha demethylase	0.0013–0.0026 µM	[43]
ENMD-2076	Serine/threonine-protein kinase	7.4–14.8 µM	[43]
Plant derivatives			
*Hyptis ovalifolia*/Argentilactone	Isocitrate lyase	4.5–36 μg/mL	[47]
*Eugenia uniflora*/oenothein B	Beta-glucana sintase	125–500 μg/mL	[70]
*Schinus terebinthifolius*/Schinol	ND	7.5–125 μg/mL	[71]
*Baccharis dracunculifolia*/Caryophyllene oxide	ND	125–250 μg/mL	[52]
*Annona cornifolia*	ND	2–250 µg/mL	[72]
*Copaifera langsdorffii*	ND	62.5 µg/mL	[37]
Fungal derivatives			
*Candida albicans*/Farnesol	Cytoplasmic degeneration	30 µM	[58]
*Aspergillus felis*/Cytochalasin E	Cell wall	7.2 µM	[55]

MFC—minimum fungicidal concentration; ND—not determined.

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
