# Peer review of "Overview of Antifungal Drugs against Paracoccidioidomycosis: How Do We Start, Where Are We, and Where Are We Going?"

_jof, 2020, doi:10.3390/jof6040300_

Round 1

Reviewer 1 Report

A brief summary: The aim of the study was to discuss new drug candidates for treatment of paracoccidioidomycosis, neglected tropical disease with limited antifungal treatment options. The authors focused on strategies used to develop and evaluate this drugs, especially those acting on pathways absent in humans. 

Broad comments: Question addressed is well defined and the results from different studies are presented and interpreted adequately. Data are presented and analyzed appropriately. Review of research methods included also the novel methods as computational modelling and nanotechnology.  The paper will attract a wide mycology public but especially those dealing with this problematic issue. The advance of publishing this work is presentation of all the latest data in this field.  

Specific comments: Figures in this paper are comprehensive and presentable.

Author Response

Thank you for sharing the comments made by the referees, which we have carefully analyzed. Their comments were very helpful and constructive and allowed us to significantly improve the manuscript. Overall, we appreciate the sympathy for our work as manifested throughout the referee’s comments and expect that the responses below have helped to clarify the potential problems raised about our work.

Point 1: English language and style are fine/minor spell check required.

Response 1:The English was reviewed.The revision certificate is attached

Reviewer 2 Report

General comments

I disagree that PCM therapy is a challenge today. Itraconazole is the drug of choice for almost all patients with mild to moderate clinical forms. Optionally, patients can also be treated with the second generations triazoles; i.e. voriconazole or isavuconazole or even with cotrimoxazole. For life treating disease, the lipid formulations of amphotericin B can be safety used as in cryptococcosis, disseminated histoplasmosis and other systemic endemic mycoses.

Specific Comments

Abstract

I don’t think that there are limitations related to the antifungal therapy in patients with PCM. Paracoccidioides spp. differs from other pathogenic fungi because it is a very sensitive organism, when exposed to antifungal drugs; even the sulfonamides can inhibit its growth.   Paracoccidioides spp. is sensitive to most of the antifungal drugs, including the triazole compounds: itraconazole, fluconazole, voriconazole Posaconazole and isavuconazole, as well as terbinafine and amphotericin B. To date, the in vitro or in vivo resistance of this fungus has not been convincingly demonstrated. So, a large therapeutic armamentarium is available to treat patients of PCM.

Introduction

Lines 41-44 As all endemic systemic mycoses, including histoplasmosis, coccidioidomycosis, talaromycosis, blastomycosis and emergomycosis, the duration of therapy is long and according to the severity of the clinical for and the immune status of the host, it may take months to years. Most of the patients with mild to moderate clinical forms respond to 6-12 months courses of itraconazole, 200 mg per day. Itraconazole is a well-tolerated drug and its cost may range from U$20 cents to U$1 dollar per capsule, depending on the country and generic versus reference brand.

Lines 47 – 61 -. The literature search to evaluate antifungal drugs studied in vitro or in vivo against Paracoccidioides spp, did not capture the new triazole compounds like posaconazole and isavuconazole. Both drugs are very active and safe against the fungus.

Lines 134-137. Once again, I think that we have a very efficiency, safe and available antifungal armamentarium to treat patients with PCM

Author Response

General comments

I disagree that PCM therapy is a challenge today. Itraconazole is the drug of choice for almost all patients with mild to moderate clinical forms. Optionally, patients can also be treated with the second generations triazoles; i.e. voriconazole or isavuconazole or even with cotrimoxazole. For life treating disease, the lipid formulations of amphotericin B can be safety used as in cryptococcosis, disseminated histoplasmosis and other systemic endemic mycoses.

Specific Comments

Abstract

Point 1. I don’t think that there are limitations related to the antifungal therapy in patients with PCM. Paracoccidioides spp. differs from other pathogenic fungi because it is a very sensitive organism, when exposed to antifungal drugs; even the sulfonamides can inhibit its growth.   Paracoccidioides spp. is sensitive to most of the antifungal drugs, including the triazole compounds: itraconazole, fluconazole, voriconazole Posaconazole and isavuconazole, as well as terbinafine and amphotericin B. To date, the in vitro or in vivo resistance of this fungus has not been convincingly demonstrated. So, a large therapeutic armamentarium is available to treat patients of PCM.

Firstly, we would like to thank the reviewer for his/her valuable comments and we believe that those comments will help to improve our review. We have changed the sentences where we indicate that the antifungal therapy against PCM is limited. The main down point of the PCM treatment is the long duration of the treatment and the side effects caused by the clinically used drugs against PCM. New treatment would be interesting in reducing this long treatment period and the side effects improving the quality of life of patients.

On line 17, the sentence “Paracoccidioidomycosis is a neglected disease that causes great economic and social impact, affecting mainly people of certain social segments, such as rural workers. Treatment is still a challenge, being carried out for years through the administration of antifungals that have several side effects and are associated with high toxicity. Due to limitations related to antifungal therapy, researchers have concentrated efforts on the identification of new compounds using numerous tools and targeting important proteins from Paracoccidioides, with emphasis on pathways absent in humans. The review aims to discuss aspects related to the identified compounds, applied methodologies and perspectives in this process of proposing new antifungal agents” was changed to “Paracoccidioidomycosis is a neglected disease that causes great economic and social impact, affecting mainly people of certain social segments, such as rural workers. Treatment is still a challenge, lasting for years through the administration of antifungals that have several side effects and are associated with high toxicity. Researchers have concentrated efforts on the identification of new compounds using numerous tools and targeting important proteins from Paracoccidioides, with the emphasis on enzymatic pathways absent in humans. This review aims to discuss the aspects related to identification of compounds, methodologies and perspectives when proposing new antifungal agents against PCM”.

Introduction

Point 2: Lines 41-44 As all endemic systemic mycoses, including histoplasmosis, coccidioidomycosis, talaromycosis, blastomycosis and emergomycosis, the duration of therapy is long and according to the severity of the clinical for and the immune status of the host, it may take months to years. Most of the patients with mild to moderate clinical forms respond to 6-12 months courses of itraconazole, 200 mg per day. Itraconazole is a well-tolerated drug and its cost may range from U$20 cents to U$1 dollar per capsule, depending on the country and generic versus reference brand.

We would like to thank the reviewer for his/her valuable comments and we believe that those comments will help to improve our review. In recent years, research that evaluates patients' adherence to the treatment of chronic diseases has shown that patients with greater resources, such as education and income, adhere better to medical recommendations [1]. Treatment with itraconazole for 6 to 12 months, which costs approximately US $ 1 per capsule, is a limiting economic factor, mainly in under developed countries where health care systems face several economic issues. The text in Lines 41-44 was improved accordingly.

“A clear concern in the treatment of PCM and other systemic mycosis is the long time needed to reduce the fungal burden and treatment might last from months to years of antifungal administration. In addition, therapeutic options are limited to the antifungals that act on two main targets, plasma membrane and folic acid synthesis [2]. Most of the patients with mild to moderate clinical forms of PCM respond well upon itraconazole treatment but, the cost of the treatment can be a limiting economic factor, mainly in under developed countries where health care systems face several economic issues. Several studies have proposed different therapeutic approaches, including more economic forms of treatment. Thus, this review focus on researches that identified alternative compounds to the current treatment of PCM, as well as strategies that used for the development of such compounds into new antifungal drugs.”

Point 3: Lines 47 – 61 -. The literature search to evaluate antifungal drugs studied in vitro or in vivo against Paracoccidioides spp, did not capture the new triazole compounds like posaconazole and isavuconazole. Both drugs are very active and safe against the fungus.

We would like to thank the reviewer for his/her valuable comments and we believe that those comments will help to improve our review. Posoconazole and Isavuconazole were added in the table supplementary 1.

In the Line 268, the phrase “The consensus also points to the possibility of using voriconazole, posaconazole and isavuconazole to replace itraconazole, paying attention to costs, clinical evidence and drug interactions” was added.

Point 4: Lines 134-137. Once again, I think that we have a very efficiency, safe and available antifungal armamentarium to treat patients with PCM.

We would like to thank the reviewer for his/her valuable comments and we believe that those comments will help to improve our review. However, as stated throughout in the manuscript, in recent years, researchers have focused on finding and developing new compounds that could be used against PCM aiming to find a shorter period of treatment, with less side effect and more economically available even if the current PCM treatment is considered efficient. This could improve the adherence of the patient to the PCM treatment.

This part of the manuscript was clarified on lines 283-284.

“The results reinforce the need for the development of alternative treatments, less toxic and economically more accessible.”

References

  1. DiMatteo, M.R. Variations in Patients??? Adherence to Medical Recommendations: A Quantitative Review of 50 Years of Research. Medical Care 2004, 42, 200–209, doi:10.1097/01.mlr.0000114908.90348.f9.
  2. Shikanai-Yasuda, M.A. Paracoccidioidomycosis treatment. Revista do Instituto de Medicina Tropical de São Paulo 2015, 57, 31–37, doi:10.1590/S0036-46652015000700007.

Reviewer 3 Report

The authors of the manuscript entitled “Overview of antifungal drugs against Paracoccidioidomycosis: How do we start, where are we and where are we going?” aimed to provide an overview of the antifungal drugs available in the treatment of Paracoccidioides infections. Globally, the work is very updated and of general interest for the journal readers. There are, however, some issues, especially regarding language and manuscript organization that the authors should address in this review.

For example:

Line 46 - “…can be used for the development…”

Line 47 – Please replace “… suggested against…” by “… used in the treatment of…”

Line 131 – Please replace “These factors call attention…” by “The results reinforced the need…”

In section 4, the most important compounds with fungicidal activity against Paracoccidioidomycosis should be resumed in a table. The others should be listed in the supplement material as stated in line 247.

The authors did not mention the role of nikkomycin z in the management of Paracoccidioides infections. A brief description should be added to the manuscript.

Author Response

Their comments were very helpful and constructive and allowed us to significantly improve the manuscript. Overall, we appreciate the sympathy for our work as manifested throughout the referee’s comments and expect that the responses below have helped to clarify the potential problems raised about our work.

Thank you for sharing the comments made by the referees, which we have carefully analyzed. Their comments were very helpful and constructive and allowed us to significantly improve the manuscript. Overall, we appreciate the sympathy for our work as manifested throughout the referee’s comments and expect that the responses below have helped to clarify the potential problems raised about our work.

Reviewer: The authors of the manuscript entitled“Overview of antifungal drugs against Paracoccidioido mycosis: How do we start, where are we and where are we going?”aimed to provide an overview of the antifungal drugs available in the treatment of Paracoccidioides infections. Globally, the work is very updated and of general interest for the journal readers. There are, however, some issues, especially regarding language and manuscript organization that the authors should address in this review.

Pont 1: Line 46-“...can be used for the development...”

Response 1: In the Line 44, the phrase “Thus, this review focus on research that identified alternative compounds to current treatments of PCM, as well as strategies that have been used to development such antifungal drugs” was changed to “Thus, this review focus on researches that identified alternative compounds to the current treatment of PCM, as well as strategies that used for the development of such compounds into antifungal drugs..”

Point 2: Line 47 Please replace “... suggested against...” by “... used in the treatment of...”

Response 2:In the Line 63, the phrase “In order to review the antifungal compounds suggested against Paracoccidioides spp. a search was carried out at SciELOwas changed to “To review antifungal compoundsused in the treatment of PCMa search was performed in SciELO.

Point 3: Line 131 Please replace “These factors call attention...” by “The results reinforced the need...”

Response 3: In the Line 282, the phraseThese factors call attention to the need for the development of research, seeking alternatives for treatment that are more efficient, less toxic and economically more accessible” was changed to “The results reinforce the need for the development of alternative treatments that are more efficient, less toxic and economically more accessible..”

Point 4: In section 4, the most important compounds with fungicidal activity against Paracoccidioido mycosis should be resumed in a table. The others should be listed in the supplement material as stated in line 247.

Response 4: In the section 4, the compounds with fungicidal activity were resumed in the table 1.

Point 5: The authors did not mention the role of nikkomycin z in the management of Paracoccidioides infections. A brief description should be added to the manuscript.

Response 5. In the section 10, Line3158, the role of nikkomycin Z in the management of Paracoccidioides infections was mentioned. The phrase “Although some of the patents are not specific to Paracoccidioides spp., the inventors of the patents suggested their usein the treatment of PCM,such as nikkomycin Z,a modified-nucleoside analog and inhibitors of chitin synthesis of dimorphic fungal pathogens.Thus, since the cell wall is an important virulence factor of Paracoccidioides spp. and absent in human cells, this compound can be a therapeutic alternative in the treatment of PCM.” was added.

Round 2

Reviewer 2 Report

Author's Notes

Regarding The main down point of the PCM treatment is the long duration of the treatment and the side effects caused by the clinically used drugs against PCM. New treatment would be interestinhig in reducing this long treatment period and the side effects improving the quality of life of patients.

Reviewer’s comment

There is no short treatment for chronic granulomatous infectious diseases like tuberculosis, toxoplasmosis and all endemic systemic mycoses, including PCM. The residual forms like the pulmonary sequela and fibrosis are the main challenge in the therapy of this mycoses. A new way to explore is how to avoid scarring and fibrosis. Pentoxiphilin showed potential benefit in experimental PCM, but it was not tested in humans

I disagree that PCM therapy is a challenge today. Itraconazole is the drug of choice for almost all patients with mild to moderate clinical forms. Optionally, patients can also be treated with the second generations triazoles; i.e. voriconazole or isavuconazole or even with cotrimoxazole. For life treating disease, the lipid formulations of amphotericin B can be safety used as in cryptococcosis, disseminated histoplasmosis and other systemic endemic mycoses.

Abstract

Differently from mycetoma and chromoblastomycosis, the only officially recognized NTD by the WHO, PCM is not a challenged to be treated. The vast majority of patients are treated with itraconazole or cotrimoxazole which are well tolerated drugs even during long course therapy. Less than 10% of the patients treated with itraconazole will present abdominal complains or transitory elevation of the hepatic enzymes. So, only amphotericin B in Deoxicholate, which is used in a minority percentage of individuals with severe clinical forms, may be associated with high toxicity.

Specific comments

Lines 88 – 89 Amphotericin B is available in Brazil in three different formulations: Deoxicholate, in lipid Complex and in liposomal. Please revise your phrase.

Lines 104-107 The main reason for the oral ketoconazole discontinuation, is its endocrinotoxicity. Ketoconazole presented the same problem as itraconazole in terms of drug-to-drug interactions and gastrointestinal absorption

Lines 122-123 I do not agree that itraconazole is unavailability in the public health system in Brazil. Itraconazole is freely provided by the Brazilian Ministry of Health and now is part of the list of drugs distributed by the RENAME (Relação Nacional de Medicamentos Essenciais), attached.

Lines 39 – 40. The only two mycoses officially recognized by the WHO are chromoblastomycosis and mycetoma Please check https://www.who.int/neglected_diseases/Ending-the-neglect-to-attain-the-SDGs--NTD-Roadmap.pdf?ua=1. PCM, sporotrichosis and other endemic mycosis are in process of recognition but not accepted yet.

Author Response

Thank you for your comments and suggestions. We have carefully analyzed them. They were very helpful and constructive and allowed us to improve the manuscript significantly.

Point 1. There is no short treatment for chronic granulomatous infectious diseases like tuberculosis, toxoplasmosis and all endemic systemic mycoses, including PCM. The residual forms like the pulmonary sequela and fibrosis are the main challenge in the therapy of this mycoses. A new way to explore is how to avoid scarring and fibrosis. Pentoxiphilin showed potential benefit in experimental PCM, but it was not tested in humans.

Response: “In the Line 128, Page 3, was added the phrase “In addition to antifungal therapy, the treatment of possible sequelae of PCM such as pulmonary fibrosis and the prevention of opportunistic diseases should also be considered. An additional therapy proposal for PCM aimed at reducing pulmonary fibrosis is the combination of itraconazole-pentoxifylline. However, although the combination has shown promising results in mice, there are still no reports on testing in humans”.

Point 2: I disagree that PCM therapy is a challenge today. Itraconazole is the drug of choice for almost all patients with mild to moderate clinical forms. Optionally, patients can also be treated with the second generations triazoles; i.e. voriconazole or isavuconazole or even with cotrimoxazole. For life treating disease, the lipid formulations of amphotericin B can be safety used as in cryptococcosis, disseminated histoplasmosis and other systemic endemic mycoses.

Response: In the abstract, Line 17, Page 1, the idea that PCM therapy is a challenge was reformulated.

Point 3: Lines 88 – 89 Amphotericin B is available in Brazil in three different formulations: Deoxicholate, in lipid Complex and in liposomal. Please revise your phrase.

Response: In the Line, 84, Page 2, the text was reformulated to “New formulations of AmB were developed as the incorporation in liposomes, resulting in better tissue distribution and less toxicity.”

Point 4: Lines 104-107 The main reason for the oral ketoconazole discontinuation, is its endocrinotoxicity. Ketoconazole presented the same problem as itraconazole in terms of drug-to-drug interactions and gastrointestinal absorption

Response: In the Line, 97, Page 3, the text about oral ketoconazole discontinuation was reformulated to “Despite the success of this drug in controlling mild and moderate forms of the disease, its use is no longer recommended due to hepatotoxicity and and

adrenal insufficiency [22]. Thus, ketoconazole was eventually replaced by the first generation of triazoles, especially after the introduction of itraconazole (ITZ) in 1987 [23].”

Point 5: Lines 122-123 I do not agree that itraconazole is unavailability in the public health system in Brazil. Itraconazole is freely provided by the Brazilian Ministry of Health and now is part of the list of drugs distributed by the RENAME (Relação Nacional de Medicamentos Essenciais), attached.

Response: In the Line 109, Page 3, the text was reformulated to “The publication of the consensus on PCM in 2006, allowed the creation of guidelines to formalize the PCM clinical treatment. Shikanai-Yasuda recommends the use of ITZ as the drug of choice for the treatment of mild and moderate forms of PCM, followed by CMX and AmB, according to the severity of the disease. The consensus also points to the possibility of using voriconazole, posaconazole and isavuconazole to replace itraconazole, paying attention to costs, clinical evidence and drug interactions [9,15].

Point 6: Lines 39 – 40. The only two mycoses officially recognized by the WHO are chromoblastomycosis               and                mycetoma                Please                                                   check https://www.who.int/neglected_diseases/Ending-the-neglect-to-attain-the-SDGs--NTD- Roadmap.pdf?ua=1. PCM, sporotrichosis and other endemic mycosis are in process of recognition but not accepted yet.

Response: In the Line 38, the text was reformulated to “The eradication of the fungus in the tissues is slow and treatment might last from months to years of antifungal administration. Paracoccidioides spp. are sensitive to various systemic antifungals, but the therapeutic options are limited to the antifungals that act on two main targets, plasma membrane and folic acid synthesis [5]. Several new compounds with antifungal properties have been proposed against PCM over the last decade. Thus, this review focus on researches that identified alternative compounds to the current treatment of PCM, as well as strategies that used for the development of such compounds into new antifungal drugs.”